# Cellular tolerance at the μ-opioid receptor is phosphorylation dependent

**Seksiri Arttamangkul, Daniel A Heinz, James R Bunzow, Xianqiang Song, John T Williams\***

The Vollum Institute, Oregon Health and Science University, Oregon, United States

**Abstract** Phosphorylation of the μ-opioid receptor (MOR) is known as a key step in desensitization and internalization but the role in the development of long-term tolerance at the cellular level is not known. Viral expression of wild type (exWT) and mutant MORs, where all phosphorylation sites on the C-terminus (Total Phosphorylation Deficient (TPD)) were mutated to alanine, were examined in locus coeruleus neurons in a MOR knockout rat. Both receptors activated potassium conductance similar to endogenous receptors in wild type animals. The exWT receptors, like endogenous receptors, acutely desensitized, internalized and, after chronic morphine treatment, displayed signs of tolerance. However, TPD receptors did not desensitize or internalize with agonist treatment. In addition the TPD receptors did not develop cellular tolerance following chronic morphine treatment. Thus C-terminal phosphorylation is necessary for the expression of acute desensitization, trafficking and one sign of long-term tolerance to morphine at the cellular level.

DOI: https://doi.org/10.7554/eLife.34989.001

## Introduction

Considerable effort has been aimed at characterizing the mechanisms that underlie acute μ-opioid receptor (MOR) dependent desensitization and cellular tolerance (*Williams et al., 2013*). One key step thought to be important in these processes is the phosphorylation of sites on the cytoplasmic loops and C-terminal tail following receptor activation. There are 11 possible phosphorylation sites on the C-terminal tail of the MOR (*Figure 1*). Two specific cassettes, amino acid residues 354 to 357 (TSST) and 375 to 379 (STANT), are phosphorylated following application of agonists that induce desensitization and internalization (*Lau et al., 2011*). Point mutations of serine (S) and threonine (T) residues in the STANT sequence resulted in a decrease in agonist induced arrestin recruitment and internalization, but did not eliminate the induction of acute desensitization (*Lau et al., 2011*; *Birdsong et al., 2015*). Complete alanine mutation of both the TSST and STANT sequences significantly reduced, but did not completely eliminate acute desensitization. Although the TSST and STANT sequences are known to be agonist-dependent phosphorylation sites, there are four other serine and threonine sites on the C-terminus that are either phosphorylated constitutively or by agonist dependent kinases (*Williams et al., 2013*). It is not known if phosphorylation of these sites alters the regulation of MORs.

The present study investigated the role of C-terminus MOR phosphorylation on acute signaling, desensitization, internalization and cellular tolerance in rats. The desensitization of MOR in rat locus coeruleus neurons measured using the activation of potassium (GIRK) conductance has been studied extensively in wild type animals. In order to study wild type and mutant MORs, a MOR-knockout rat model was used and MORs were virally expressed. Two versions of MOR were expressed, wild-type (exWT) and total phosphorylation deficient receptors (TPD) where all 11 possible phosphorylation sites on the C-terminus were mutated to alanine. By linking green fluorescence protein (GFP) to the N-terminus of the MOR the trafficking of expressed receptors was examined. Plasma membrane

**\*For correspondence:**
williamj@ohsu.edu

**Competing interests:** The authors declare that no competing interests exist.

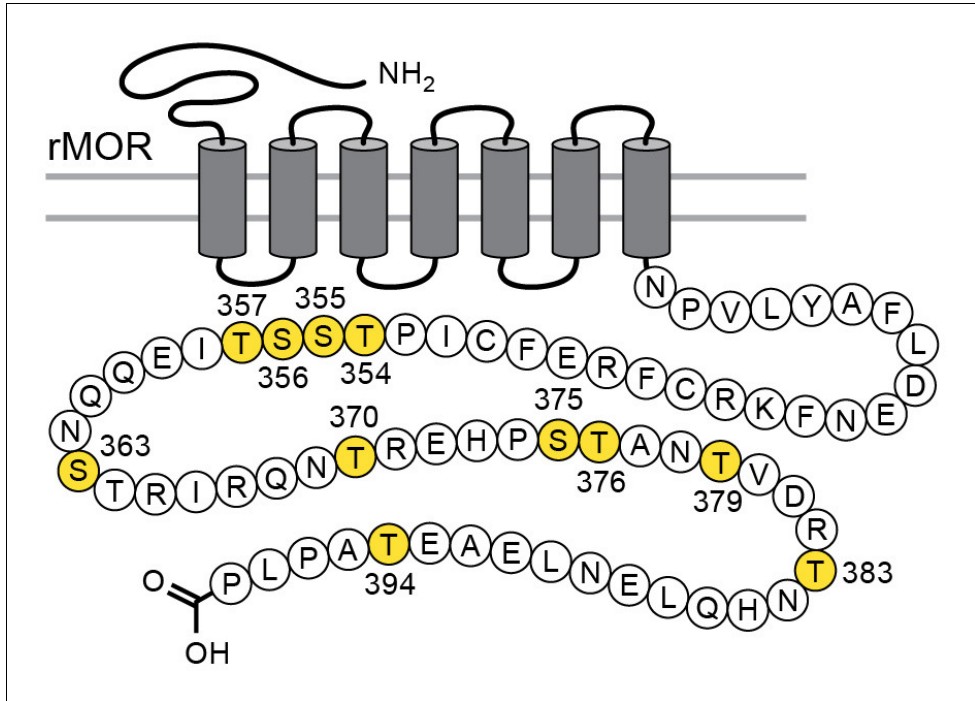

**Figure 1.** Schematic illustrates the phosphorylation sites (in yellow) that were mutated to alanine for the TPD receptor.
DOI: https://doi.org/10.7554/eLife.34989.002

associated receptors were selectively labeled with the use of a fluorescently tagged antiGFP nano-body. The results show that both receptors activate a potassium current that hyperpolarizes the cells. There was no significant difference in the kinetics of activation between WT, exWT, and TPDs. Thus acute activation of virally expressed MORs was not significantly different from endogenous receptors. There was however a near complete loss of desensitization, internalization, and long-term tolerance in neurons expressing TPD receptors. This study demonstrates a key role of phosphoryla-tion in the both acute- and long-term actions of opioids on the activation of potassium conductance on single neurons.

# Results

## MOR knockout

Recordings from LC neurons in brain slices from the MOR knockout animal confirmed that there was no opioid–induced current or hyperpolarization. The activation of alpha-2-adrenoceptors, orphanin FQ, or M3-muscarine receptors however acted as in slices taken from wild type animals (*Figure 2*, *Figure 2—figure supplement 1*).

## MOR expression

Microinjections of adeno-associated virus type two containing either the exWT or TPD receptors were made bilaterally into the LC and after 2–4 weeks the GFP tagged MORs were visualized with an Olympus macroview microscope (*Figure 3A*). Images obtained with a 2-photon microscope showed green fluorescence on the plasma membrane and in the cytoplasm of LC neurons. Plasma membrane receptors were selectively targeted and labeled using an anti-GFP nanobody conjugated to alexa594 dye (*Figure 3B–D*). There was no labeling of cells from knockout animals that did not express GFP tagged MORs (*Figure 3E,F*).

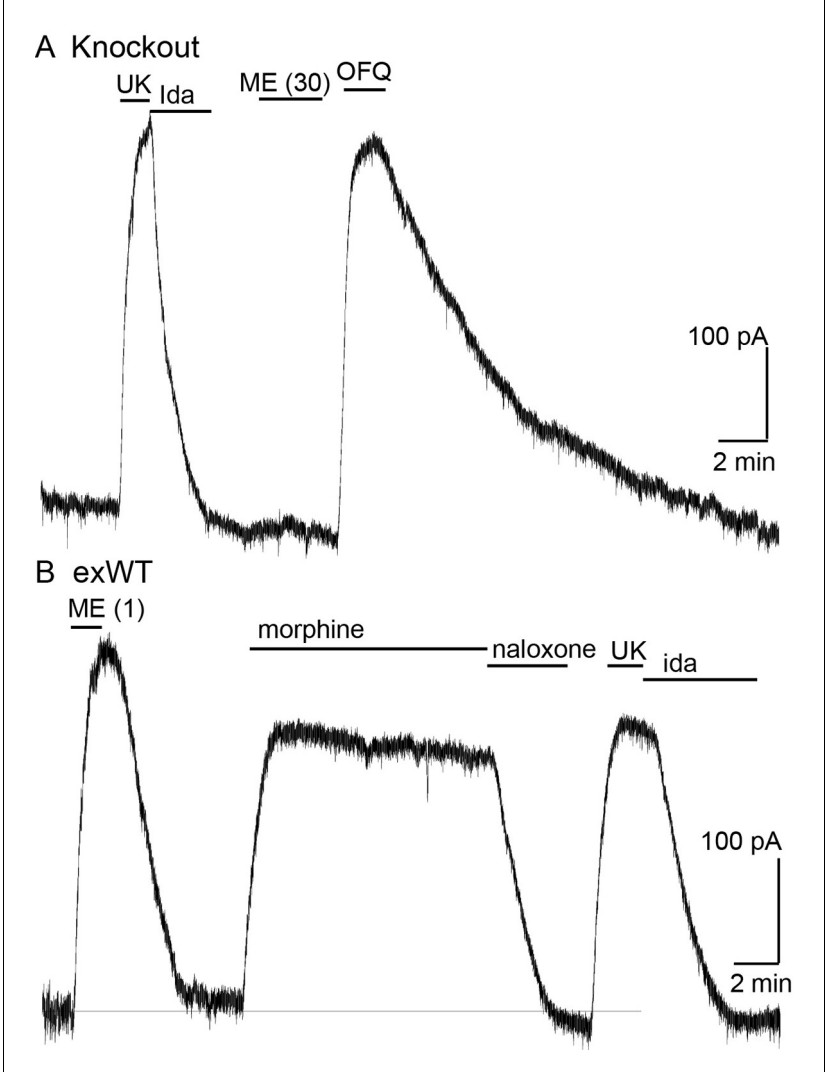

**Figure 2.** Locus coeruleus neurons are not sensitive to opioids in the MOR knockout rat. (**A**) from a MOR knockout animal where the alpha-2-adrenoceptor agonist, UK14304 (3 µM) and OFQ both activate potassium currents whereas ME (30 µM) had no effect. (**B**) from a neuron where the wild type MOR (exWT) was expressed in the MOR knockout animal. In this recording ME (1 µM), morphine (10 µM) as well as UK14304 (3 µM) all caused outward currents.

DOI: https://doi.org/10.7554/eLife.34989.003

The following figure supplement is available for figure 2:

**Figure supplement 1.** The firing rate of locus coeruleus neurons is not changed by opioids in recordings from the MOR knockout animal.

DOI: https://doi.org/10.7554/eLife.34989.004

## Electrophysiology of the virally expressed receptors

Recordings were made from slices from adult male and female animals 2–4 weeks following viral microinjection. Each preparation was examined with the macroview microscope to roughly determine the level of receptor expression. Preparations were hemisected and whole-cell voltage clamp or intracellular membrane potential recordings were made from LC neurons. Often times the expression was not the same on both sides of the preparation. The application of [Met[5]]enkephalin (ME, 300 nM, an EC50 concentration in slices from wild type animals) was used as an estimate of the level of receptor expression. In preparations from wild type rats ME (300 nM) results in an outward current between 100 and 250 pA or 10–20 mV hyperpolarization (*Quillinan et al., 2011*; *Levitt and*

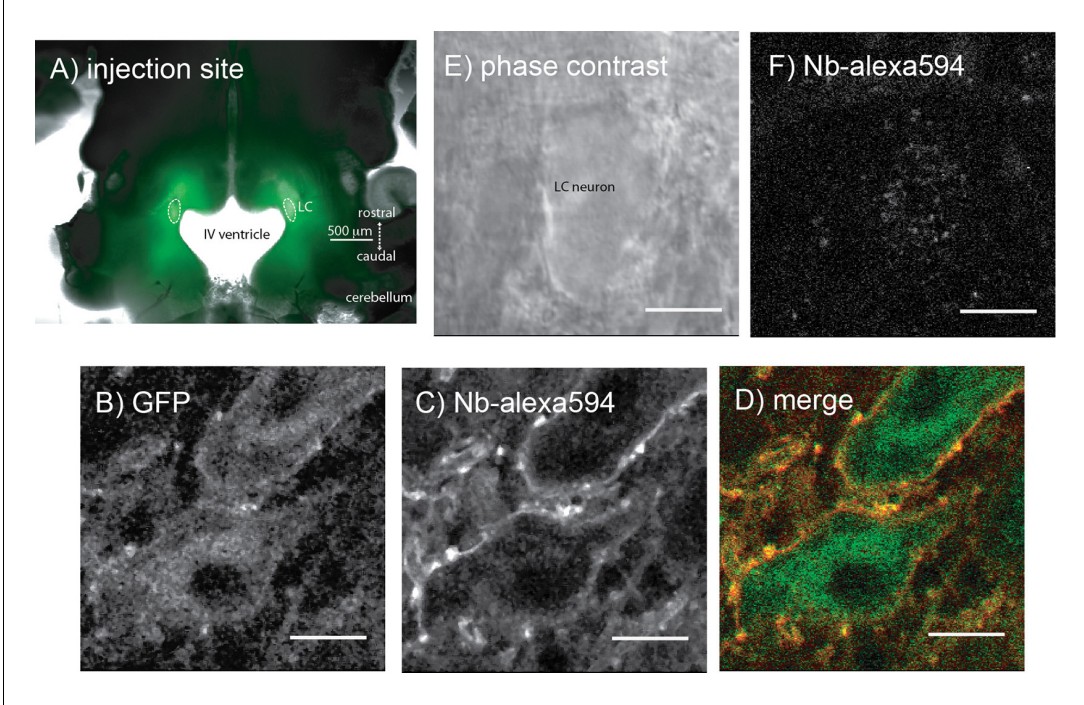

**Figure 3.** Microinjection of virus expressing GFP-tagged exWT receptors in the locus coeruleus of MOR knockout rat. (**A**) low power image of the GFP fluorescence observed in a horizontal slice containing the LC. (**B**) shows an image obtained with a 2-photon microscope showing the GFP fluorescence. (**C**) the same neuron showing the plasma membrane associated receptors following incubation of the slices with an anti-GFP nanobody conjugated with alexa594. (**D**) the merged image of the GFP and alexa594 fluorescence. (**E**) a scanning DIC image of a LC neuron from a knockout animal that had not been injected with virus. (**F**) the same neuron following incubation of the slice with the anti-GFP nanobody conjugated with alexa594 showing that without the expression of the GFP-tagged receptors there was no staining. Scale bar = 10 μm.
DOI: https://doi.org/10.7554/eLife.34989.005

*Williams, 2012*; *Dang and Williams, 2004*; *Fiorillo and Williams, 1996*). Preparations were not included in the study if the fluorescence was low and the outward current or hyperpolarization induced by the application of ME (300 nM) was less than 30 pA or 5 mV, respectively.

The kinetics of receptor-dependent activation of G protein-gated inwardly rectifying potassium (GIRK) conductance was examined with the photolysis of caged-enkephalin (CYLE) using whole-cell recordings in the voltage clamp configuration (*Figure 4*). The rate of activation (10–90%, exWT 255 ± 35 ms, n = 10, TPD 220 ± 32 ms, n = 14, p=0.5, unpaired T-test) and the time to the peak of the outward current (exWT 1.78 ± 0.12 s, n = 12, TPD 1.95 ± 0.18, n = 17, p=0.4, unpaired T-test) were the same between exWT and TPD and similar to those from experiments made from cells expressing wild type receptors (*Williams, 2014*). Likewise the return to baseline following photolysis of the caged antagonist naloxone (CNV-Nal) in the presence of ME (1 μM) was not different in recordings made from the two receptors (exWT 1.57 ± 0.08 s, n = 7, TPD 1.53 ± 0.15, n = 6, p=0.78, unpaired T-test). The decay of the ME current induced by CNV-Nal was thought to result from processes downstream of the receptor, most likely dependent on the sequestration of G protein beta/gamma subunits away from the GIRK channels (*Banghart et al., 2013*). The time course of current decay induced by high affinity agonists was found to be a better indication of agonist/receptor interaction. Therefore experiments were also done with the high affinity agonist endomorphin-1 (100 nM). The time constant of CNV-Nal-induced inactivation in exWT receptors (4.06 ± 0.41 s, n = 8) was not significantly different from that measured with TPD receptors (5.37 ± 0.59 s, n = 10, p=0.09. unpaired T-test). The current amplitude induced by endomorphin-1 was also not different (290 ± 37 pA in exWT, n = 8, and 255 ± 44 pA in TPD, n = 10, p=0.55, unpaired T-test). Thus, in spite of the multiple mutations along the C-terminus and the presence of GFP at the N-terminus, the acute

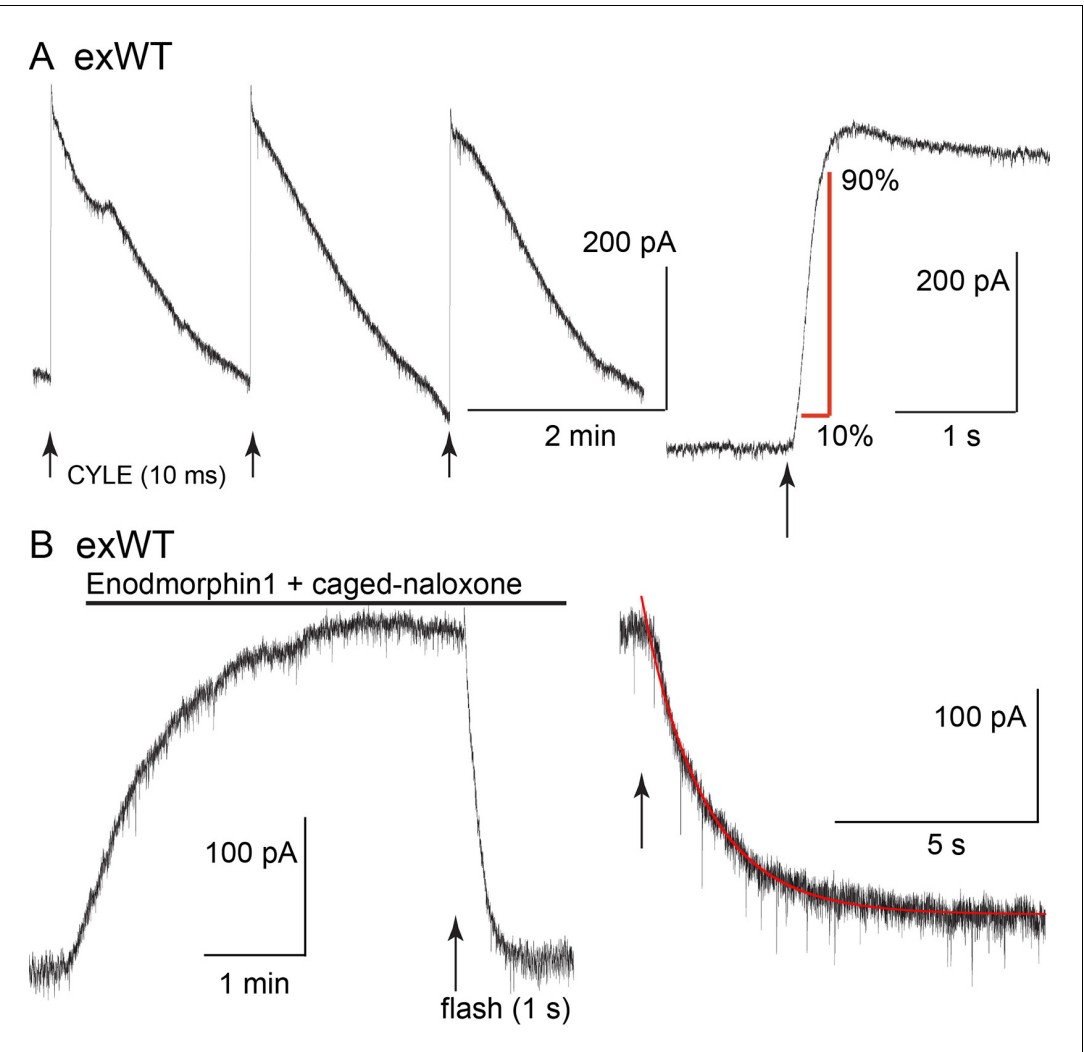

**Figure 4.** The kinetics of activation and inhibition of expressed wild type receptors (exWT) in LC neurons from the MOR knockout measured with photolysis of caged enkephalin (CYLE). (**A**) repeated photolysis events (1/2 min) resulted in rapidly rising outward currents. Right side, the rate of rise was measured as the time it took to go from 10% to 90% of the peak. (**B**) the high affinity agonist endomorphin1 (100 nM) was applied along with caged-naloxone (5 μM). At the arrow a 1 s photolysis flash was applied and the decrease in the outward current was measured. Right side is an expanded time base of the decrease in outward current and the single exponential fit to that decline.

DOI: https://doi.org/10.7554/eLife.34989.006

signaling of virally expressed TPD and exWT receptors was not significantly different from each other or receptors from wild type animals.

## Desensitization

Acute receptor desensitization was compared in slices from animals injected with either exWT or TPD receptors. Two measures of desensitization were made. One was the decrease in the peak (hyperpolarization using intracellular recordings or current with whole-cell recordings) during the application of a saturating concentration of ME (30 μM, 10 min). The second was the relative decrease in response to a sub-saturating concentration of ME (300 nM) following the washout of the saturating concentration of ME (*Figure 5*, *Figure 5—figure supplement 1*). The results from experiments with exWT receptors were similar to results from wild type animals that have been published previously (*Quillinan et al., 2011*). During the application of ME (30 μM) the hyperpolarization of

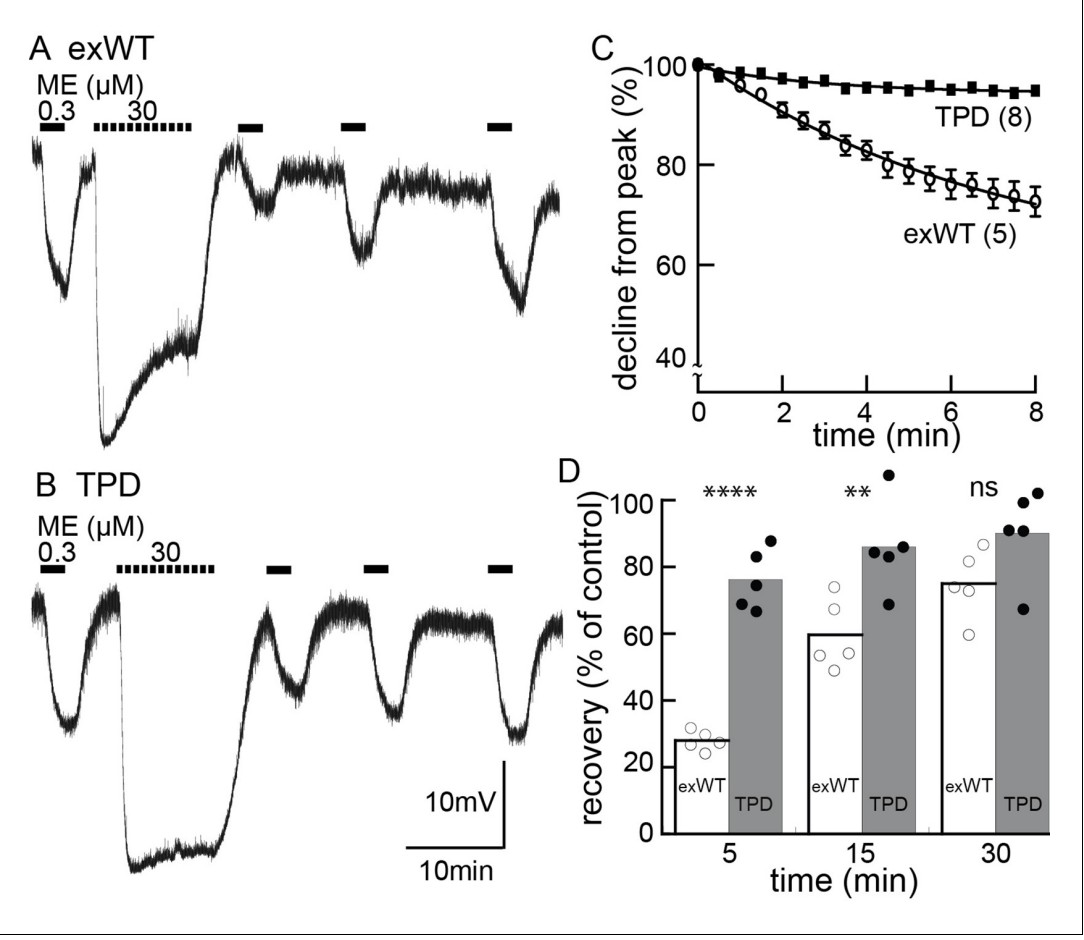

**Figure 5.** Desensitization is largely blocked in experiments carried out in neurons expressing the TPD. Experiments carried out with intracellular recordings of membrane potential. (**A**) an example of an experiment with a cell expressing the exWT receptor. ME (300 nM) was applied before and following application of a saturating concentration of ME. The amplitude of the hyperpolarization induced by ME (30 μM) decreased during the 10 min application. The hyperpolarization induced by ME (300 nM) was reduced and recovered slowly following washout of ME (30 μM). (**B**) the same experiment carried out in neurons that expressed the TPD receptor. (**C**) summarized results showing the decline in the hyperpolarization during the application of ME (30 μM). There is only a small decline in experiments carried out with the TPD receptor. (**D**) summarized results showing the recovery from desensitization. Experiments with neuron in wild type animal (WT) and the expressed exWT receptors show a slow recovery, whereas there was little sign of desensitization in the experiments from the TPD receptor.
DOI: https://doi.org/10.7554/eLife.34989.007

The following figure supplement is available for figure 5:

**Figure supplement 1.** Reduced desensitization of the TPD receptor measured with whole cell voltage clamp recording.
DOI: https://doi.org/10.7554/eLife.34989.008

membrane potential measured with intracellular recording declined to 75 ± 5% of the peak (n = 6). The current measured under voltage clamp with whole-cell recording declined to 54 ± 4% of the peak (n = 12). Following the washout of saturating ME (30 μM), the hyperpolarization induced by ME (300 nM) was reduced to 30 ± 2% (n = 5) and the current to 23 ± 5% (n = 10) of the pre-desensitization response. Following a 20–30 min wash of ME (30 μM) the application of ME (300 nM) approached the pre-desensitization control with whole cell voltage clamp or intracellular recordings.

The results obtained with the TPD receptors were significantly different from the exWT receptors. With intracellular recordings, the hyperpolarization induced by ME (30 μM, 10 min) declined to 94 ± 1% of the peak (n = 7, compared with exWT experiments, 75 ± 5%, n = 6, p=0.0004, unpaired

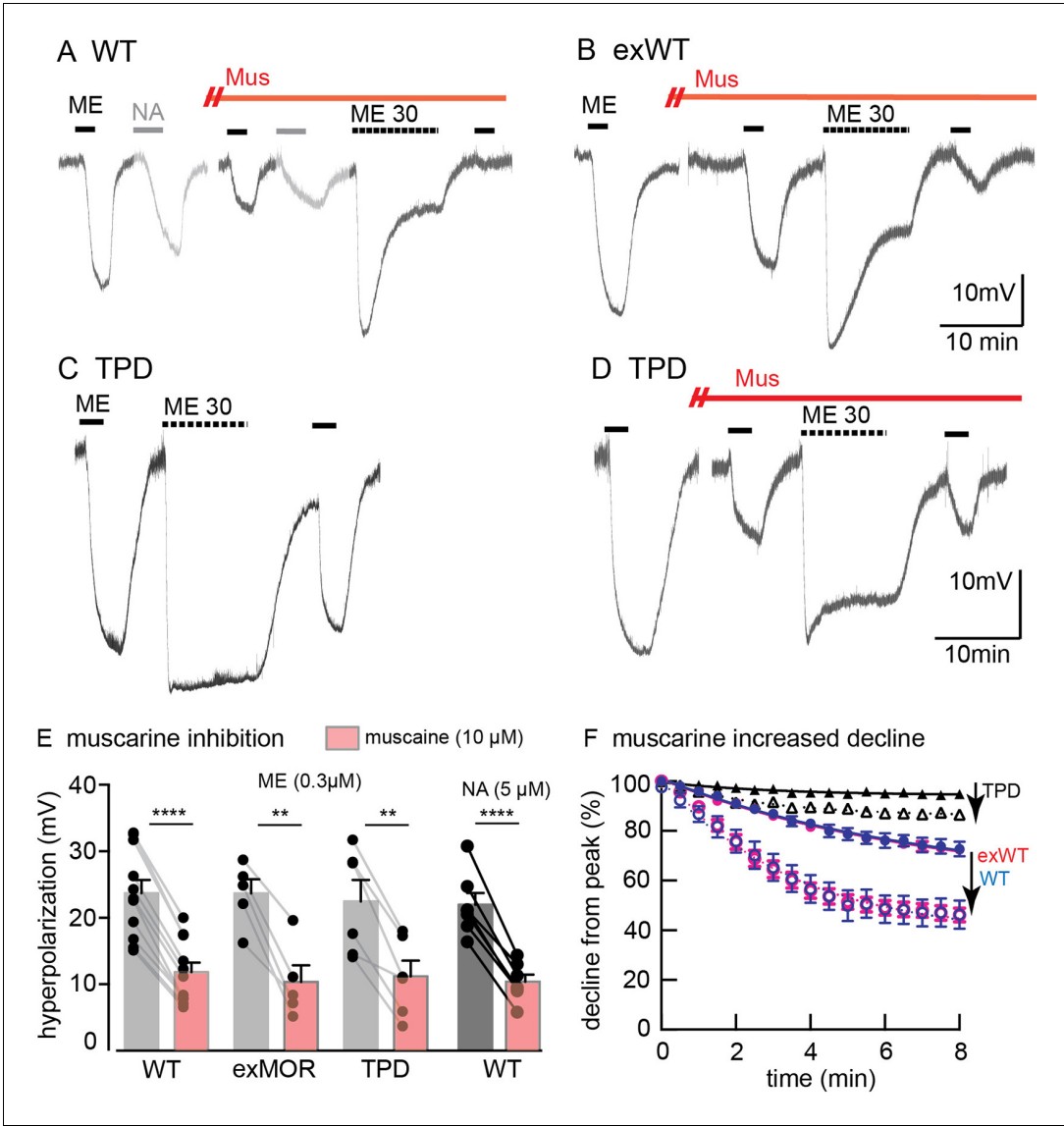

**Figure 6.** Muscarine inhibits ME-induced hyperpolarization and increases desensitization of WT, exWT and TPD receptors. Recording of membrane potential made using intracellular electrodes. (**A**) a recording from a neuron in a slice from a wild type animal. The hyperpolarizations induced by ME (300 nM) and noradrenline (NA 5 μM, plus cocaine 3 μ M to prevent reuptake and prazosin 500 nM to block alpha-1-adrenoceptors) were both decreased in the presence of muscarine (10 μM). The hyperpolarization induced by ME (30 μM) peaked and declined during the 10 min application. (**B**) the same experiment carried out in a recording from an exWT receptor. The hyperpolarization induced by ME (300 nM) was reduced and the decline in the hyperpolarization induced by ME (30 μM) was increased in the presence of muscarine (10 μM). (**C**) an experiment with the TPD receptor showing the lack of decline in the hyperpolarization during the application of ME (30 μM). (**D**) the same experiment carried out in the presence of muscarine (10 μM). The initial hyperpolarization induced by ME (300 nM) is reduced and there is a greater decline in the hyperpolarization during the application of ME (30 μM). (**E**) summary of the inhibition of the initial hyperpolarization induced by ME and NA in WT, exWT and TPD receptors, two-tailed paired t-test (**p<0.01, ****p<0.0001). (**F**) summary of the decline in the hyperpolarization in control and in the presence of muscarine. The muscarine-induced increase in decline in experiments from WT and exWT receptors is the same (WT n = 15 in control 11 in muscarine; exWT n = 5 in control, five in muscarine). There is also a small but significant increase in the decline found in experiments with the TPD receptor (n = 8 in control, six in muscarine, p<0.0001, two way ANOVA, Bonferroni post hoc).

DOI: https://doi.org/10.7554/eLife.34989.009

The following figure supplement is available for figure 6:

*Figure 6 continued on next page*

*Figure 6 continued*
**Figure supplement 1.** Whole cell voltage clamp experiment showing the inhibition of the outward current induced by photolysis of CYLE caused by muscarine (10 µM).
DOI: https://doi.org/10.7554/eLife.34989.010

T-test). With whole cell recordings the current declined to 80 ± 2% of the peak (n = 13, compared with exWT experiments, 54 ± 4%, n = 12, p<0.0001, unpaired T-test). Likewise there was only a small decrease in the response to ME (300 nM) 5 min following the washout of the saturating concentration of ME (intracellular, TPD 76 ± 4%, n = 5, compared to exWT 28 ± 1%, n = 5, p<0.0001, one-way ANOVA, Bonferroni post hoc, *Figure 5*; whole cell, TPD 81 ± 9%, n = 11, exWT 23 ± 5%, n = 10, p<0.0001, unpaired T-test). Thus by two measures obtained using two recording conditions, acute desensitization was markedly reduced in neurons expressing TPD receptors.

## Protein kinase C dependent desensitization

MOR desensitization induced by protein kinase C (PKC) has been proposed to result from phosphorylation at S363, T370 or S/T356-357 (*Wang et al., 2002*; *Feng et al., 2011*; *Chen et al., 2013*; *Illing et al., 2014*). Receptor desensitization was augmented by the PKC activators, phorbol 12,13-dibutyrate and phorbol 12-myristate 13-acetate that increased the activity of PKC in wild type animals. Muscarine, also thought to activate PKC by a Gq-dependent mechanism, induced a large increase in apparent desensitization (*Arttamangkul et al., 2015*). Muscarine facilitated the desensitization induced by ME (30 µM) with intracellular recordings from neurons expressing exWT or TPD receptors (*Figure 6*). In the presence of muscarine, the initial amplitude of the hyperpolarization induced by ME (300 nM) decreased by 43 ± 8% in experiments from exWT, n = 5 and by 49 ± 8% from TPD, n = 6. The decline from the peak hyperpolarization induced by ME (30 µM, 10 min) was also facilitated (*Figure 6*). Although the decline from the peak in experiments examining the TPD receptors was small, it was more than double that measured in the absence of muscarine (6 ± 1%, n = 7 *vs.* 15 ± 2% in muscarine, n = 6, p=0.0009, unpaired T-test). Likewise, in whole-cell voltage clamp experiments, the current amplitude evoked by photolysis of CYLE was significantly decreased by muscarine and returned to baseline following the application of the muscarinic antagonist scopolamine (*Figure 6—figure supplement 1*). Thus the presumed activation of PKC induced by muscarine acts by a mechanism that is independent of phosphorylation of the C-terminal of MOR.

## Receptor trafficking – acute and following chronic morphine treatment

Receptor internalization was studied by immuno-labeling the extracellular N-terminal GFP of plasma membrane-associated exWT and TPD receptors. Living slices were incubated in an anti-GFP nanobody-alexa594 for 30–45 min, placed in a superfusion chamber and visualized with 2-photon microscopy (*Figure 7*). Labeled receptors were imaged before and after the application of a saturating concentration of ME (30 µM, 10 min) resulting in the internalization of exWT receptors (*Figure 7A* top). When TPD receptors were examined using the same treatment protocol there was no detectable internalization (*Figure 7A* bottom). Thus, ME induced internalization of TPD MORs was completely disrupted (exWT receptor internalization was 38 ± 3% of the total fluorescence, n = 7; internalization of the TPD was 10 ± 5% of the total fluorescence, n = 6, p=0.0004, unpaired T-test).

The development of long-term tolerance induced by chronic morphine treatment after expression of exWT and TPD receptors was examined next. A previous study in mouse LC found that the recycling of FLAG-MORs was increased after chronic morphine treatment in the arrestin3 knockout animals (*Quillinan et al., 2011*). It is possible that chronic morphine treatment could modulate the trafficking of expressed exWT and TPD receptors. Animals were microinjected with virus to express either exWT or TPD and after 7–10 days treated with morphine (80 mg/kg/day) using osmotic mini pumps. After 6 or 7 days brain slices were prepared in morphine-free solutions for imaging with a 2-photon microscope. As expected, exWT receptors were internalized in slices from morphine treated animals (49 ± 6%, n = 4, *Figure 7B* top). There was however, no internalization of TPD receptors in slices from morphine treated animals (0%, n = 4, *Figure 7B* bottom, p=0.002, unpaired T-test). Thus the inability to induce internalization of TPD receptors was maintained following chronic morphine treatment.

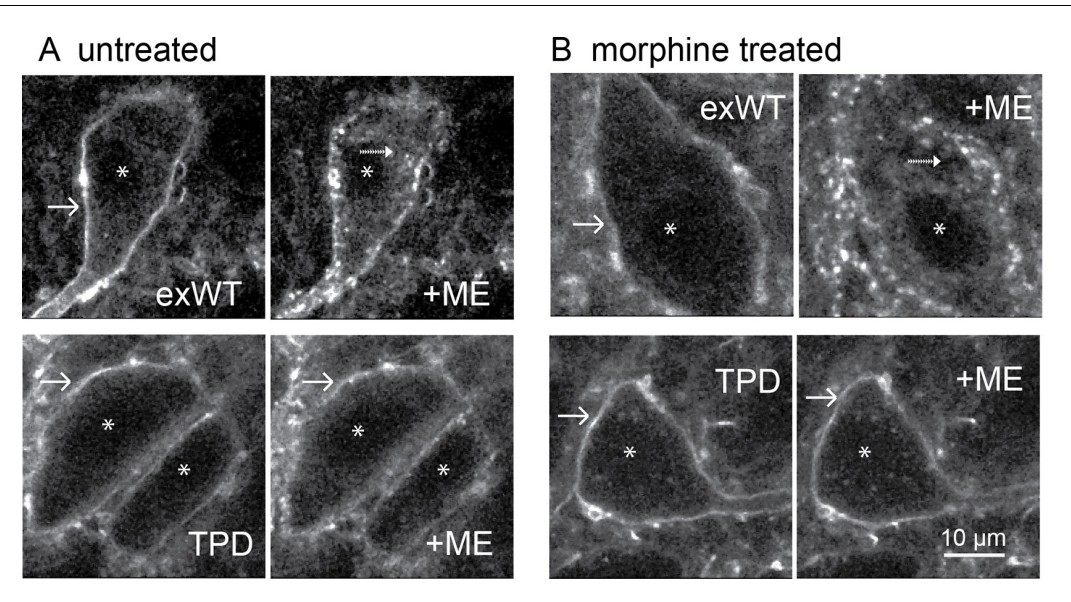

**Figure 7.** Receptor trafficking of the TPD is blocked in slices from untreated and morphine treated animals. Images from exWT (top) and TPD (bottom) showed receptor distribution before and following application of ME (30 μM, 10 min). The exWT receptors became punctate and moved into the cytoplasm in slices from untreated (A, top) and morphine treated (B, top) animals. The TPD receptors did not traffic in slices from untreated (A, bottom) or morphine treated animals (B, bottom). Scale bar = 10 μm. Solid-arrows showed plasma membrane receptors. Dash-arrows showed receptors that were internalized after ME treatment. Asterisks indicated nucleuses.

DOI: https://doi.org/10.7554/eLife.34989.011

## Tolerance following chronic morphine treatment

Whole-cell voltage clamp experiments using slices from morphine treated animals were used to examine the development of tolerance. One limitation of the use of virus is variability in the level of receptor expression from one experiment to the next. The relative current induced by low (ME 300 nM) compared to the high concentration of ME (30 μM) varied widely (*Figure 8—figure supplement 1*). This problem limited the ability to construct concentration response curves taken from slices from untreated and morphine treated animals to be used as a measure of tolerance.

As an alternative, the extent and recovery from acute desensitization in slices from morphine treated animals was examined in animals with viral expression of receptors. This was a robust measure of tolerance found in wild type animals where acute desensitization was increased and the recovery from desensitization was delayed (*Dang and Williams, 2004*; *Quillinan et al., 2011*). Desensitization was greater in slices from morphine treated animals expressing exWT MORs than in untreated controls. First, the current that remained at the end of a 10 min application of ME (30 μM) was from untreated animals was 59 ± 4% of peak (n = 8), significantly larger than in slices from morphine treated animals (41 ± 2% of peak, n = 8, p=0.001, unpaired T-test). Second, the current induced by the application of ME (300 nM) 5 min following the washout of ME (30 μM) in slices from morphine treated animals was significantly depressed relative to the control (untreated 35 ± 6%, n = 8; MTA 13 ± 2%, n = 6, p=0.01, unpaired T-test, *Figure 8B*). As was reported in experiments using wild type animals, following washout of the ME (30 μM) the recovery of the current induced by ME (300 nM) in slices expressing exWT receptors from morphine treated animals was also significantly reduced (*Figure 8*, p=0.0001, two way ANOVA). Thus acute desensitization and the time course of recovery from desensitization of the exWT receptors indicated the development of tolerance.

In experiments from morphine treated animals expressing the TPD receptor, there was no significant change in any measure of opioid action (*Figure 8A,C*). The decline from the peak current induced by a saturating concentration of ME (30 μM 10 min) was 81 ± 3% (n = 9) and not different from experiments in slices taken from untreated animals (84 ± 2%, n = 10, p=0.34, unpaired T-test).

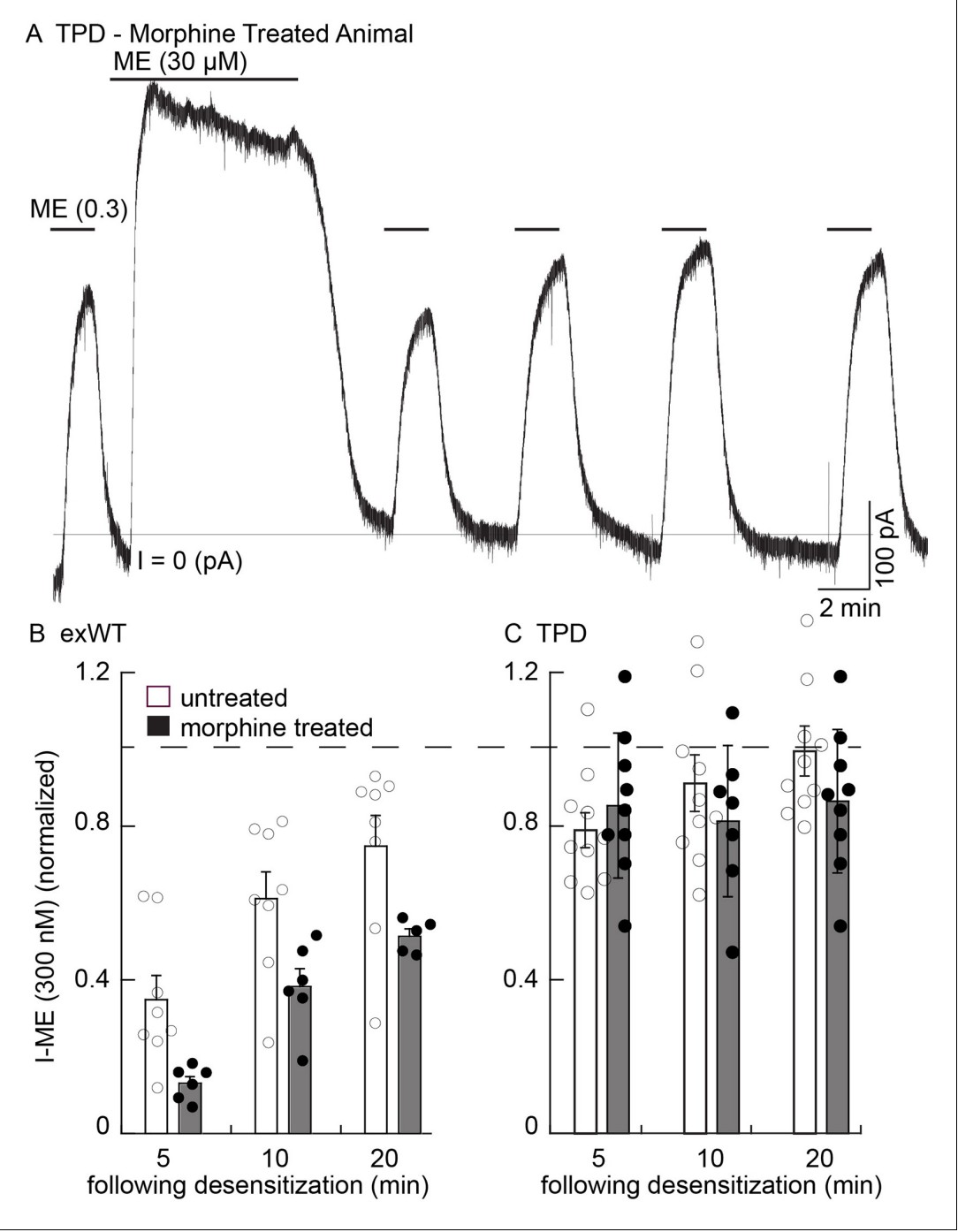

**Figure 8.** There is no sign of one measure of long-term tolerance induced in the TPD receptor following chronic morphine treatment. Whole cell voltage clamp experiments. (**A**) an example from a morphine treated animal expressing the TPD receptor. There was a small decline in the current induced by a saturating concentration of ME (30 μM, 10 min) and a small and transient decrease in the current induced by ME (0.3 μM) following the washout of the ME (30 μM). (**B**) summary shows the recovery from desensitization induced by ME (30 μM, 10 min) in exWT. There is significantly less recovery from desensitization seen in experiments from morphine treated animals than untreated controls with the exWT receptor (p=0.0001, two way ANOVA, Bonferroni post hoc). (**C**) summarized results from untreated and the morphine treated animals expressing the TPD receptors. There was a small (not significant) decrease in the ME (0.3 μM) current immediately after the ME (30 μM) treatment in experiments from animals expressing the TPD receptor that did not change with time. There were no difference with repeated applications of ME (300 nM) in slices from untreated and morphine treated animals at any point.

*Figure 8 continued on next page*

*Figure 8 continued*

DOI: https://doi.org/10.7554/eLife.34989.012

The following figure supplement is available for figure 8:

**Figure supplement 1.** Variation in receptor expression limits the ability to construct concentration response curves among groups of animals.

DOI: https://doi.org/10.7554/eLife.34989.013

The current induced by ME (300 nM) at all times following the washout of the ME (30 μM) solution was also not different from slices of untreated animals (untreated, n = 10; MTA, n = 9, two way ANOVA, *Figure 8C*).

Finally, the initial rise in current induced by photolysis of CYLE was not different in slices from untreated and morphine treated animals (10–90% rise time of the first flash, untreated exWT 255 ± 35 ms (n = 10), TPD 219 ± 32 ms (n = 14); morphine treated exWT 201 ± 21 ms (n = 10), TPD 211 ± 23 (n = 10)). Thus the rising phase of the current was not changed following chronic morphine treatment (p=1, ANOVA, Bonferroni post hoc).

Desensitization was also examined in slices from morphine treated animals using photolysis of caged-enkephalin (CYLE, *Figure 9*). The peak current and 10–90% rise time was measured in response to repeated 10 ms flashes before and after a longer (100 ms) flash. In experiments from exWT expressing cells the amplitude of the current declined significantly by the fifth flash (87 ± 3% of the first flash, n = 13, p=0.002, ANOVA Dunnett post hoc). Following the long flash, there was a step decrease in the peak current measured in the exWT receptors on the 7th and 11th flash (seventh flash 69 ± 3% of the first flash, n = 13, p<0.0001, Dunnett post hoc, *Figure 9C*). Likewise the rise time was significantly slowed (the fifth flash = 126 ± 4% of the first flash, p=0.005, ANOVA, Dunnett post hoc, *Figure 9D*). In contrast, the long flash did not alter the peak amplitude or activation rate of the 10 ms flash-induced currents in recordings from cells expressing TPD receptors (n = 9, ANOVA, Dunnett post hoc, *Figure 9D*). Thus, the normalized peak currents and activation rates following the long flash were significantly different between exWT and TPD receptors further indicating that TPD receptors display little to no desensitization even in slices from morphine treated animals.

## Discussion

The acute activation, desensitization, endocytosis and the development of tolerance to opioids were characterized in locus coeruleus neurons from a MOR knockout rat following the viral expression of exWT and TPD MORs. The results with the exWT receptors in the knockout animal were similar to those found in wild type animals. The activation kinetics of TPD receptors was also the same as both virally expressed and endogenous WT receptors. This study, measuring the increase in potassium conductance, demonstrates that the elimination of phosphorylation sites along the C-terminal of the MOR largely eliminates acute desensitization and one measure of long-term cellular tolerance to morphine. The results also indicate that with the expression of receptors specifically in the locus coeruleus the development of tolerance is cell autonomous, independent of receptor activation in other areas of the CNS.

### Desensitization and internalization

MOR desensitization and internalization are not closely linked. The strongest evidence is based on experiments using mutation of the STANT sequence. Although mutations in the STANT sequence resulted in a significant decrease in internalization, acute desensitization was little changed (*Lau et al., 2011*; *Just et al., 2013*; *Birdsong et al., 2015*). The conclusion was that acute desensitization precedes internalization and the two processes are mechanistically distinct. The same conclusion was reached in experiments in cultured locus coeruleus neurons from a transgenic mouse that expressed soluble GFP under the control of the tyrosine hydroxylase promotor (TH-GFP). Internalization of the endogenous MORs measured with the use of a fluorescent peptide, dermorphin-Alexa594, was blocked by concanavalin A and yet desensitization and the recovery from desensitization measured electrophysiologically was not changed (*Arttamangkul et al., 2006*). The sequence beginning at T354 and ending at T357 (TSST) was also efficiently phosphorylated following the

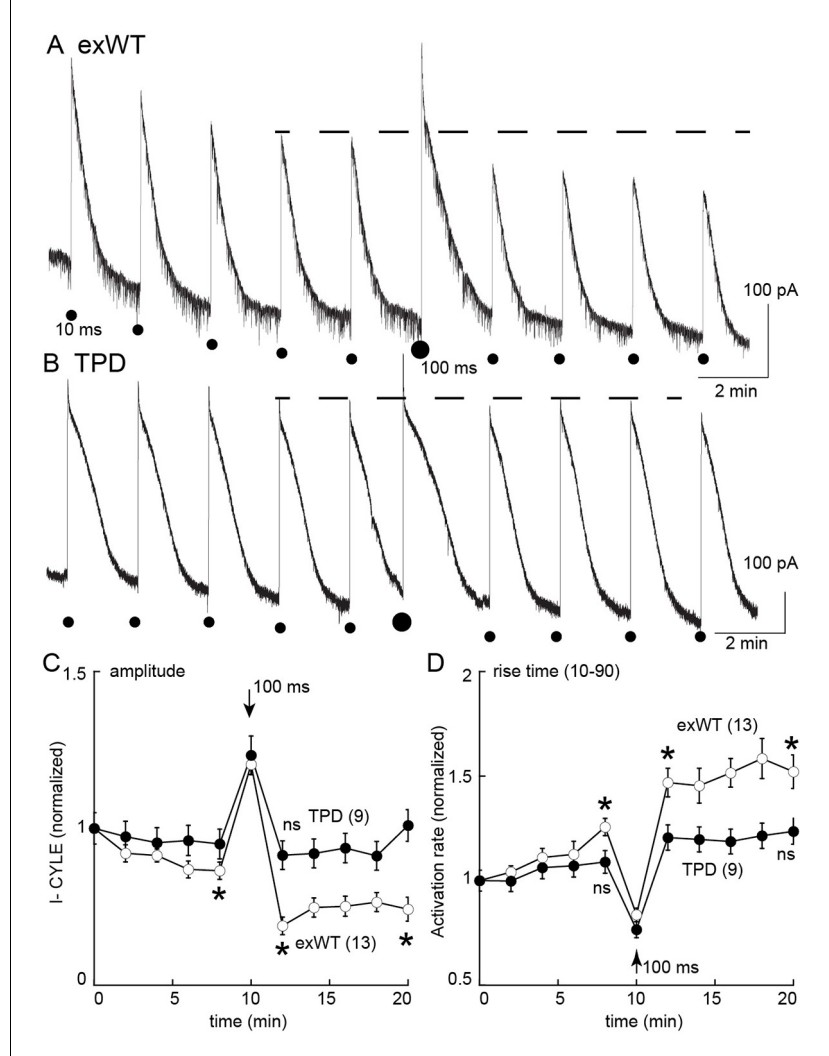

**Figure 9.** Measures of acute desensitization induced by photolysis of caged enkephalin (CYLE) in slices from morphine treated animals. Whole cell voltage clamp recordings were made from neurons expressing the TPD or exWT receptors. (**A**) exWT and (**B**) TPD are example experiments where photolysis of CYLE was carried out every 2 min. The initial duration of the flash was 10 ms, after five flashes the duration of the flash was increased to 100 ms and subsequent flashes were 10 ms. (**C**) shows summarized results of the amplitude of the outward currents. In recordings from exWT receptors the amplitude of the current induce by the 10 ms flash decreased steadily. Increasing the duration of the flash increased the peak current and the amplitude of the subsequent currents induced by 10 ms flashes was decreased. The current induced by the 10 ms flashes in experiments from the TPD receptor were not changed (ANOVA, Dunnett test). (**D**) Summary of the rise time (10–90%) of the outward current in exWT and TPD receptors. The rise increased steadily in exWT receptors whereas there was no significant change in recordings from the TPD receptors.

DOI: https://doi.org/10.7554/eLife.34989.014

application of potent opioid agonists (*Lau et al., 2011*). An alanine-mutant of C-terminal phosphory-lation sites excluding TSST and T394 (6S/T-A sites) prevented receptor endocytosis while desensiti-zation remained unchanged (*Yousuf et al., 2015*). It was not until the TSST sequence along with the STANT sequence or all phosphosites on the C terminus were mutated to alanine that there was a large decrease in the degree of acute desensitization (*Birdsong et al., 2015*; *Yousuf et al., 2015*). Similar results were shown in this study with viral expressed TPD in MOR knockout rats. Thus it is clear that although acute desensitization and internalization of MOR are dependent on

phosphorylation, the two processes involve mechanisms that can be distinguished by the degree of phosphorylation of the C-terminal tail.

The heterologous desensitization induced by muscarinic receptor activation was not blocked after mutations of C-terminal phosphorylation sites. Activation of muscarinic receptors can increase PKC activity that is thought to underlie heterologous desensitization of MORs (*Bailey et al., 2004*). The decrease in opioid current induced by muscarine in slices expressing the TPD was similar to results obtained in slices from wild type animals (*Shen and North, 1992*; *Fiorillo and Williams, 1996*). The results may be interpreted in two ways. One is that the facilitation of MOR desensitization by PKC results from phosphorylation of intracellular loops of MOR and not in the C-terminus (*Chen et al., 2013*). It is also possible that phosphorylation of other signaling proteins results in the inhibition (*Chu et al., 2010*). Activation of muscarinic receptors has been shown to alter the trafficking of MORs heterologously expressed in HEK293 cells (*Lopez-Gimenez et al., 2017*), however there was no change in internalization of FLAG-MORs in the mouse locus coeruleus (*Arttamangkul et al., 2015*).

## Is the TPD a G protein biased receptor?

Depending on the agonist, MORs can activate differential downstream processes. A number of biased agonists have been described recently (*Siuda et al., 2017*). G protein biased agonists have reduced ability to recruit arrestin while maintaining signaling through G proteins. Mutations in the C-terminus of MORs have a significant impact on arrestin signaling. Mutations of even one phosphorylation site at S375 and/or the STANT sequence in the C-terminus resulted in a significant reduction in the association with arrestin and the induction of internalization (*Lau et al., 2011*). The STANT mutant MOR was functional as measured by adenylyl cyclase (*Lau et al., 2011*) and activation of potassium conductance (*Birdsong et al., 2015*) assays. Mutations of all phosphorylation sites on the C-terminus (TPD) eliminated endocytosis induced by highly efficacious agonists such as DAMGO, etonitazene and fentanyl in HEK293 cells (*Just et al., 2013*) as well as ME in LC neurons (present study). Given these observations, the TPD receptor could be considered to function almost exclusively through a G protein-dependent process.

## Cellular tolerance and homeostatic mechanisms

One robust measure of tolerance in the LC is a decrease in the recovery from acute desensitization in slices from morphine treated animals (*Dang and Williams, 2004*; *Quillinan et al., 2011*; *Levitt and Williams, 2012*). In experiments from slices expressing TPD receptors, chronic morphine treatment had no effect. There are two possible conclusions that follow this observation. First, the near complete block of desensitization alone reduced tolerance. This however requires further investigation using a longer duration of morphine treatment. It is possible that tolerance may develop slower. It is also possible that compensatory homeostatic mechanisms due to continuous signaling may develop from the longer-term chronic treatment. The second conclusion centers on the elimination of internalization. It may be that internalization mediated by the arrestin pathway is a key step in the development of long-term tolerance. Studies using arrestin3-null mice show that cellular tolerance is attenuated (*Quillinan et al., 2011*; *Dang et al., 2011*; *Connor et al., 2015*). This could be the result of MOR trafficking because while the internalization of MORs in these animals was normal, receptor recycling was faster after chronic morphine treatment (*Quillinan et al., 2011*). Interpretation of these studies is difficult given that only one isoform of arrestin was knocked out. Experiments expressing phosphorylation mutants in the C-terminus in the MOR knockout rat could begin to address the role of internalization in the development of cellular tolerance.

### Summary

The present work addresses one mechanism that underlies the development of long-term tolerance to morphine. Phosphorylation of the C-terminal of the MOR has been shown to be a key step in the mechanism of acute desensitization measured with multiple downstream effectors. Elimination of phosphorylation sites on the C-terminal rendered MORs resistant to one cellular measure of long-term tolerance induced by morphine. One conclusion is that desensitization and/or internalization of MORs is necessary for the development of this form of cellular tolerance to opioids. Although there were no obvious change in cellular excitability, it could be that continued signaling through the

phosphorylation deficient receptors result in downstream homeostatic mechanisms that counteract the lack of cellular tolerance and may well increase signs of withdrawal.

## Materials and methods

### Drugs

Morphine sulfate and morphine alkaloid were obtained from the National Institute on Drug Abuse (NIDA), Neuroscience Center (Bethesda, MD). Naloxone was purchased from Abcam (Cambridge, MA), MK-801, from Hello Bio (Princeton, NJ), UK14304 tartrate and idazoxan (Ida) from Tocris (Bio-Techne Corp. Minneapolis, MN). Potassium methanesulfonate was from Alfa Aesar (Ward Hill, MA). [Met$^5$] enkephalin (ME), endomorphin-1, muscarine, scopolamine, idazoxan and other reagents were from Sigma (St. Louis, MO). Caged-enkephalin (CYLE) and Caged-naloxone (CNV-Nal) were gifts from Mathew Banghart.

Morphine alkaloid was converted to salt form with 0.1 M HCl and made up a stock solution in water. The working solution was diluted in artificial cerebrospinal fluid (ACSF) and applied during incubation or superfusion. Naloxone, endomorphin-1, muscarine, scopolamine, UK14304 tartrate and idazoxan were dissolved in water, diluted in ACSF and applied by bath superfusion. Bath perfusion of ME was with bestatin (10 mM) and thiorphan (1 mM) to limit breakdown of ME.

### Animals

All animal experiments were conducted in accordance with the National Institutes of Health guidelines and with approval from the Institutional Animal Care and Use Committee of the Oregon Health and Science University (Portland, OR). Adult (180–300 g or 5–6 weeks) male and female Sprague-Dawley rats were obtained from Charles River Laboratories (Wilmington, MA). A pair of MOR-knockout Sprague-Dawley rats with ZFN target site (GCTGTCTGCCACCCAgtcaaaGCCCTGGATTTC within exon 2) were generated by Horizon (St. Louis, MO) and received as F3 generation. The animals were bred and raised in house for two more generations before used in the experiments. The gene deletion was confirmed by genotyping using the primer 5'CATATTCACCCTCTGCACCA3'.

### Microinjection protocol

MOR-knockout animals (24–30 days) were anesthetized with isofluorane (Terrell, Piramal Clinical Care, Inc., Bethlehem, PA) and placed in a stereotaxic frame for micro-injection of viral particles containing adeno associated virus type two for the expression of wild type MORs (exWT, AAV2-CAG-SS-GFP-MOR-WT-WPRE-SV40pA, 2.06 x $10^{13}$ vg/ml) and total phosphorylation deficient MORs (TPD, AAV2-CAG-SS-GFP-MOR-TPD-WPRE-SV40pA, 2.21 × $10^{13}$ vg/ml). Both viruses were obtained from Virovek (Hayward, CA). Injections of 200 nl at the rate of 0.1 µl/min were done bilaterally at ±1.25 mm lateral from the midline and −9.72 mm from the bregma at a depth of 6.95 mm from the top of the skull using computer controlled stereotaxic Neurostar (Kähnerweg, Germany). Experiments were carried out 2–4 weeks following the injection.

### Animal treatment protocols

Rats (5–6 weeks) were treated with morphine sulfate continuously released from osmotic pumps as described previously (*Quillinan et al., 2011*). Osmotic pumps (2ML1, Alzet, Cupertino, CA) were filled with the required concentration of morphine sulfate in water to deliver 80 mg/kg/day. Each pump has a 2 ml reservoir that releases 10 µl/hour for up to 7 days. Rats were anesthetized with isoflurane and an incision was made in the mid-scapular region for subcutaneous implantation of osmotic pumps. Pumps remained in animals until they were used for experiments 6 or 7 days later.

### Tissue preparation

Horizontal slices containing locus coeruleus (LC) neurons were prepared as described previously (*Williams and North, 1984*). Briefly, rats were killed and the brain was removed, blocked and mounted in a vibratome chamber (VT 1200S, Leica, Nussloch, Germany). Horizontal slices (250–300 µm) were prepared in warm (34°C) artificial cerebrospinal fluid (ACSF, in mM): 126 NaCl, 2.5 KCl, 1.2 MgCl$_2$, 2.6 CaCl$_2$, 1.2 NaH$_2$PO$_4$, 11 D-glucose and 21.4 NaHCO$_3$126 and 0.01 (+) MK-801 (equilibrated with 95% O2/5% CO2, Matheson, Basking Ridge, NJ). Slices were kept in solution with (+)

MK-801 for at least 30 min and then stored in glass vials with oxygenated (95% O2/5% CO2) ACSF at 34°C until used.

## Electrophysiology

Slices were hemisected and transferred to the recording chamber which was superfused with 34°C ACSF at a rate of 1.5–2 ml/min. Whole-cell recordings were made from LC neurons with an Axo-patch-1D amplifier in voltage-clamp mode ($V_{hold}$ = −60 mV). Recording pipettes (1.7–2.1 MΩ, World Precision Instruments, Saratosa, FL) were filled with internal solution containing (in mM): 115 potassium methanesulfonate or potassium methyl sulfate, 20 KCl, 1.5 MgCl$_2$, 5 HEPES(K), 10 BAPTA, 2 Mg-ATP, 0.2 Na-GTP, pH 7.4, 275–280 mOsM. Series resistance was monitored without compensation and remained <15 MΩ for inclusion. Data were collected at 400 Hz with PowerLab (Chart version 5.4.2; AD Instruments, Colorado Springs, CO). Intracellular recordings of membrane potential were made with glass electrodes (50–80 MΩ, World Precision Instruments, Saratosa, FL,) filled with KCl (2 M) and an Axoclamp-2A amplifier. Hyperpolarizing current (<20 pA) was used to prevent spontaneous firing of LC neurons. The depolarization induced by muscarine was corrected with the addition of more hyperpolarizing current to inhibit firing. Most drugs were applied by bath superfusion. In some experiments, [Leu[5]]enkephalin was applied by photolysis of caged-[Leu[5]]enkephalin (CYLE). A solution containing CYLE (30 µM), bestatin (1 µM) and thiorphan (10 µM) was recycled for photolysis experiments. In other experiments naloxone was released from the solution of caged-naloxone (CNV-Nal, 5 µM) recycled in the presence of agonist (ME 1 µM, or endomorphin1 100 nM). Photolysis was carried out with a full-field illumination of a 365 nm LED lamp (Thorlabs, Inc., Newton, NJ) attached to the epifluorescence port.

## Anti-GFP nanobody expression and purification

A nanobody recognizing GFP was obtained from Addgene (Cambridge, MA) and cloned into the pET-22b vector with N-terminal 8xHis-tag followed by thrombin cleavage site. The lysine at 116 of nanobody was mutated to cysteine for a single dye-labeling site. Protein expression was conducted in *E-coli* strain BL21 (New England BioLabs, Ipswich, MA) in Terrific Broth medium. The cell culture was grown to OD$_{600}$0.7 to 1.0 at 37°C, and protein synthesis was induced by 0.5 mM of isopropyl β-D-1-thiogalactopyranoside and was fermented at 20°C for 18 hr. Cells were harvested by centrifugation and then lysed in a lysing buffer (in mM): 50 HEPES, 500 NaCl, 5 DL-dithiothretiol, 10 imidazol and 10% glycerol using a sonicator (10 min, 4 s sonication, 8 s paused, on ice). The debris was eliminated by centrifugation and the clear lysate was purified using the HisTrap column (GE Healthcare, Marlborough, PA). The His-tag was removed by adding thrombin protease (Sigma-Aldrich (St. Louis, MO) in to the protein solution at 1:100 (by mass) and incubated at 4°C overnight. The protein was further purified by size-exclusive chromatography (Superdex 200) in Dulbecco's Phosphate Buffered Saline (Thermo Fisher Scientific, (Waltham, MA). Peak fractions having a single band by SDS-Page (10–20% gradient) electrophoresis were pooled and concentrated to ≈0.6 mg/ml.

## Anti-GFP nanobody Alexa594 conjugation

The site-specific fluorescent labeling of the cysteine-mutated nanobody was modified from previously described (*Pleiner et al., 2015*). A solution containing the nanobody (100 µg) was used for the conjugation reaction. The solution was mixed with tris-(2-carboxyethyl)phosphine 15 mM on ice for 10 min. The buffer was exchanged to labeling buffer using P6 spin-column (BioRad, Hercules, CA). A labeling reaction was started by adding 1.5-fold of Alexa 594 maleimide (2 µl of 5 µg/µl in dimethyl-sulfoxide). The reaction proceeded on ice for 1 hr. The conjugated nanobody was further purified by Superdex 200 in phosphate buffer. The degree of labeling was determined by measuring OD at 280 and 594 and was close to 1.

## MOR-GFP trafficking

Brain slices (240 µm) from the virally injected rats were prepared as previously described. Slices were visualized with an Olympus Macroview fluorescent microscope for GFP expression in the LC area and then incubated in a solution of anti-GFP nanobody Alexa594 (Nb-A594, 10 µg/mL, 30–45 min). Images were captured with an upright microscope (Olympus, Center Valley, PA.) equipped with a custom-built two-photon apparatus and a 60x water immersion lens (Olympus LUMFI, NA1.1, Center

Valley, PA). The dye was excited at 810 nm. Data were acquired and collected using Scan Image Software ([*Pologruto et al., 2003*]. Images were taken at a magnification where a single neuron filled the field of view. A z-series of 15–20 sections was collected at 1 μm intervals. With this protocol, the whole neuron was qualitatively compared. Drugs were applied by perfusion at the rate of 1.5 ml/min. All experiments were done at 35 ˚C. The area of interest was obtained by manually drawing a line along the plasma membrane. A subset of 5 sections from the z-series was selected before and after ME application. The stacks were summed using Image J z-projection and the fluorescence within the area of the plasma membrane was measured. Internalization was calculated as the difference in cytoplasmic fluorescence before (C) and after ME application (D) and normalized to fluorescence before ME (% internalization = (D-C/C)x100).

## Data Analysis
Analysis was performed using GraphPad Prism four software (La Jolla, CA). Values are presented as mean ± SEM. Statistical comparisons were made using T-test or two-way ANOVA, as appropriate. Comparisons with $p < 0.05$ were considered significant.

## Acknowledgements
This work was supported by NIH funding DA08163 (JTW). We thank members of the Gouaux lab for help in preparing the anti-GFP nanobody and members of the Williams lab for comments on the work.

## Additional information

### Funding

| Funder | Grant reference number | Author |
| --- | --- | --- |
| National Institute on Drug Abuse | RO1-DA08163 | John T Williams |

It is true that NIDA had no role in study design, data collection and interpretation, or the decision to submit the work for publication.

### Author contributions
Seksiri Arttamangkul, John T Williams, Conceptualization, Data curation, Formal analysis, Supervision, Funding acquisition, Validation, Investigation, Methodology, Writing—original draft, Project administration, Writing—review and editing; Daniel A Heinz, Conceptualization, Resources, Data curation, Formal analysis, Supervision, Validation, Investigation, Visualization, Methodology, Writing—review and editing, Tool building, Data curation; James R Bunzow, Data curation, Formal analysis, Investigation, Data curation; Xianqiang Song, Resources, Tool building

### Author ORCIDs
Daniel A Heinz  http://orcid.org/0000-0002-9450-6242
John T Williams  https://orcid.org/0000-0002-0647-6144

### Ethics
Animal experimentation: All experiments were done in accordance with the Institutional Animal Care and Use Committees (IACUCs) at Oregon Health & Science University (OHSU), protocol number IP# 00000160. A colony of MOR knockout rats was maintained in house.

### Decision letter and Author response
Decision letter https://doi.org/10.7554/eLife.34989.017
Author response https://doi.org/10.7554/eLife.34989.018

## Additional files

**Supplementary files**
• Transparent reporting form

DOI: https://doi.org/10.7554/eLife.34989.015

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
