## [Decision Letter]

Thank you for submitting your article "Cellular tolerance at the µ-opioid receptor is phosphorylation dependent" for consideration by *eLife*. Your article has been reviewed by three peer reviewers, and the evaluation has been overseen by a Reviewing Editor and Eve Marder as the Senior Editor. The following individual involved in review of your submission has agreed to reveal his identity: Ken Mackie (Reviewer #1).

The reviewers have discussed the reviews with one another and the Reviewing Editor has drafted this decision to help you prepare a revised submission..

Summary:

This study presents new data on the role of phosphorylation of the C-terminal tail of the mu opioid receptor (MOR) to receptor activation/inhibition kinetics, desensitization, internalization, and tolerance. The model system is the MOR knockout rat with viral reconstitution of wild-type MOR or a mutant in which all 11 C-terminal phosphorylation sites are mutated. Particularly attractive features of the study include expressing in the MOR null background, careful kinetic analysis of the expressed receptor signaling, examination of both hyperpolarization and outward currents, the nano-body imaging of MOR internalization in live slices and complementing bath application of agonist by flash photolysis release of met-enkephalin. The data indicate that the residues mutated in TPD MOR are responsible for the acute desensitization of GIRK currents to met-enkephalin via MOR and the development of cellular tolerance to MOR. Overall this is a well-done study that advances our understanding of the regulation of MOR signaling and the development of tolerance.

Essential revisions:

1) The conclusion that chronic morphine didn't affect MOR TPD signaling was based on the lack of enhancement of desensitization of met-enkephalin signaling in this mutant after chronic morphine treatment. It is important to determine if chronic morphine produced tolerance to met-enkephalin activation of GIRKs. For example, in Quillinan et al., 2011, chronic morphine produced a ~2-fold right shift in the met-enkephalin dose response curve. As 300 nM met-enkephalin is close to the EC50 for this response, the authors should be able to extract from their existing data if chronic morphine suppressed the GIRK response in TPD MOR.

2) Desensitization was only examined for GIRK current activation for met-enkephalin. Thus, conclusions need to be tempered that desensitization of other pathways and ligands may involve other/additional mechanisms. For example, extensive work from the Chavkin group suggests involvement of both arrestin and JNK in tolerance to various MOR agonists.

3) The imaging data require quantifying. Information about how often the results were obtained or how robust the internalization/loss of internalization actually is should be provided. Can the authors provide some kind of nominal statement of how representative the images are – in terms of number of fields and experiments?

4) Group sizes are small in many instances; statistical analyses are either not applied, not reported, inconsistently reported, or inappropriately conducted. The basis for excluding some recordings on the basis of size of the ME-induced response (and how many cells were excluded on this basis) should be provided. Without evaluating the EC50 for ME with reconstituted receptors, it does not seem justified to suggest that any concentration used in this study is saturating or an EC50 dose; the variable application of drugs across experiments (as displayed in figures) seems quite problematic, and there seems to convey that consistency from experiment to experiment was not ensured or the assumption that it is not particularly important.

---

## [Author Response]

Essential revisions:1) The conclusion that chronic morphine didn't affect MOR TPD signaling was based on the lack of enhancement of desensitization of met-enkephalin signaling in this mutant after chronic morphine treatment. It is important to determine if chronic morphine produced tolerance to met-enkephalin activation of GIRKs. For example, in Quillinan et al., 2011, chronic morphine produced a ~2-fold right shift in the met-enkephalin dose response curve. As 300 nM met-enkephalin is close to the EC50 for this response, the authors should be able to extract from their existing data if chronic morphine suppressed the GIRK response in TPD MOR.

This is something that we struggled with. The key point is that with the viral expression of receptors there was enough variability between animals that constructing reliable concentration response curves was problematic. Given the small shift in the concentration response curves that have been published in the LC in the past (2-fold) there is little chance that a similar shift would be detectable. The plot shows the current induced by ME (0.3 µM) normalized to the current induced by ME (30 µM). The variability in the current induced by the low concentration of ME (0.3 µM) relative to the high concentration limits the ability to claim tolerance in this way.

Figure 8—figure supplement 1 points out at least one drawback associated with the viral expression of receptors.

The change in acute desensitization induced by chronic morphine treatment was a robust measure of tolerance induced by chronic morphine treatment.

2) Desensitization was only examined for GIRK current activation for met-enkephalin. Thus, conclusions need to be tempered that desensitization of other pathways and ligands may involve other/additional mechanisms. For example, extensive work from the Chavkin group suggests involvement of both arrestin and JNK in tolerance to various MOR agonists.

The text has been tempered to include comment that the present study only measured the activation of potassium conductance. Measures of recovery from desensitization require the use of agonists that wash out of the slice. These experiments are limited to the use of enkephalin. This is an important point that has been made in many previous publications.

3) The imaging data require quantifying. Information about how often the results were obtained or how robust the internalization / loss of internalization actually is should be provided. Can the authors provide some kind of nominal statement of how representative the images are – in terms of number of fields and experiments?

The text in the Materials and methods section now includes more detail on how measurements were obtained. Annotations were also included in the figure to identify plasma membrane staining, internalized puncta and nucleus.

4) Group sizes are small in many instances; statistical analyses are either not applied, not reported, inconsistently reported, or inappropriately conducted. The basis for excluding some recordings on the basis of size of the ME-induced response (and how many cells were excluded on this basis) should be provided. Without evaluating the EC50 for ME with reconstituted receptors, it does not seem justified to suggest that any concentration used in this study is saturating or an EC50 dose; the variable application of drugs across experiments (as displayed in figures) seems quite problematic, and there seems to convey that consistency from experiment to experiment was not ensured or the assumption that it is not particularly important.

The numbers of experiments and the statistics are now presented in the text and illustrated in each figure. The problems surrounding the measures of concentration response is a weakness that was unavoidable with the viral expression. It is true that the exact determination of EC50 was not rigorously determined. To carry out multiple concentrations in each recording prior to starting an experiment with each cell is simply not feasible. The plot above shows the variability in the current induced by ME 300 nM relative to the high concentration. The text has been changed to state that the ME 300 nM concentration was chosen because it is the EC50 in wild type animals.

As with all viral injections there were hits, misses or near misses that resulted in no or very low levels of expression in the LC, particularly since the area of LC is very small. The preparations where expression was predicted to be low based on examination of the fluorescence prior to recording were examined but often not included because the response to the low concentration of ME (300 nM) was small (30 pA or 5 mV). It was pointless to think that a reliable decrease in those responses could be accurately measured following desensitization. In addition, the expression could be great on one side and completely absent on the other side of the LC. No tally of misses versus hits was recorded. The text has been changed to be clear.